# No-Reference Image Quality Assessment Combining Swin-Transformer and Natural Scene Statistics

**DOI:** 10.3390/s24165221

**Published:** 2024-08-12

**Authors:** Yuxuan Yang, Zhichun Lei, Changlu Li

**Affiliations:** 1School of Microelectronics, Tianjin University, Tianjin 300072, China; changlu@tju.edu.cn; 2Institute of Sensors and Measurements, University of Applied Sciences Ruhr West, 45479 Mülheim an der Ruhr, Germany; zhichun.lei@hs-ruhrwest.de

**Keywords:** image quality assessment, multi-scale features, natural scene statistics, Swin-Transformer

## Abstract

No-reference image quality assessment aims to evaluate image quality based on human subjective perceptions. Current methods face challenges with insufficient ability to focus on global and local information simultaneously and information loss due to image resizing. To address these issues, we propose a model that combines Swin-Transformer and natural scene statistics. The model utilizes Swin-Transformer to extract multi-scale features and incorporates a feature enhancement module and deformable convolution to improve feature representation, adapting better to structural variations in images, apply dual-branch attention to focus on key areas, and align the assessment more closely with human visual perception. The Natural Scene Statistics compensates information loss caused by image resizing. Additionally, we use a normalized loss function to accelerate model convergence and enhance stability. We evaluate our model on six standard image quality assessment datasets (both synthetic and authentic), and show that our model achieves advanced results across multiple datasets. Compared to the advanced DACNN method, our model achieved Spearman rank correlation coefficients of 0.922 and 0.923 on the KADID and KonIQ datasets, respectively, representing improvements of 1.9% and 2.4% over this method. It demonstrated outstanding performance in handling both synthetic and authentic scenes.

## 1. Introduction

The rapid development of communications, digital technology, and computer science has revolutionized the way information is exchanged. Images, as a crucial medium for conveying information due to their intuitive and content-rich nature, have become one of the preferred methods for people to acquire and communicate information. In reality, billions of photos are uploaded to the internet daily, making the collection, storage, and dissemination of images a routine activity. However, these processes can lead to distortions, thereby reducing image quality. During the collection phase, images may be compromised due to poor lighting conditions or limitations of the imaging equipment; in the storage phase, to save space, images need to be compressed, leading to noticeable compression artifacts; in the dissemination phase, encoding and decoding errors may occur, causing loss of image information. These quality losses can lead to insufficient and inaccurate expression of information, affecting people’s visual experience [1,2]. For example, in medical imaging, minor changes in image quality can affect the accuracy of diagnoses; in the field of security surveillance, the clarity of images directly relates to the recognizability of events. Therefore, monitoring and enhancing image quality throughout these processing stages has become an urgent issue to address.

Image Quality Assessment (IQA) evaluates the quality of images based on visual characteristics. IQA is divided into two methods: subjective image quality assessment and objective image quality assessment. Subjective image quality assessment relies on the Human Visual System (HVS), and involves qualitative evaluations of images based on human subjective perceptions, assigning quality labels to them. These labels represent different quality levels of images, allowing for standardized quality comparisons. Using human subjective perception to assess image quality is considered reliable; however, this method requires the participation of multiple subjects and must be conducted in a specific environment, making it time-consuming and impractical for real-world applications. In contrast, objective image quality assessment develops algorithms to simulate the human eye’s perception of images, enabling the automatic assessment of image quality without human intervention, which saves time and effort [1]. This approach has achieved significant results in fields such as image restoration, medical imaging, and video surveillance.

Objective image quality assessment is generally categorized into three types based on the amount of information provided by the original reference image: Full-Reference IQA (FR-IQA), Reduced-Reference IQA (RR-IQA), and No-Reference IQA (NR-IQA); the latter is also known as Blind IQA (BIQA) [2]. FR-IQA utilizes both the original (undistorted, reference) images and the distorted images for assessment. NR-IQA operates solely with the distorted image, presenting a higher level of difficulty due to the absence of reference images. RR-IQA falls between NR-IQA and FR-IQA, providing only partial information or extracted features from the original image. Since reference images are typically unavailable in practical scenarios, NR-IQA holds substantial practical significance due to its broad applicability and ease of entation.

The essence of image quality assessment is to measure image quality using models that simulate the perceptual mechanisms of HVS. Convolutional Neural Network (CNN)-based methods excel at capturing local details in the image, but ignore the context and global information of the image, while Transformer-based deep learning models are able to focus on global details, but they frequently encounter challenges related to high computational resources and a large number of parameters. This not only increases the time cost of model training and inference, but also imposes high hardware requirements for deployment environments, limiting their application on resource-constrained devices. Furthermore, both CNN-based and Transformer-based models often involve resizing images, which can lead to the loss of important information.

To address these issues, we propose a no-reference image quality assessment model that combines Swin-Transformer and natural scene statistics (STNS-IQA). It employs the Swin-Transformer as the backbone network to extract multi-scale information from images and uses natural scene statistics methods to compensate for information loss due to image resizing. Our contributions can be summarized as follows:We propose an NR-IQA method called STNS-IQA, which employs Swin-Transformer to extract multi-scale features from images, and natural scene statistics to compensate for the information loss caused by image resizing, thus improving the performance of image quality assessment.We develop a Feature Enhancement Module (FEM) that enhances global visual information capture using dilated convolution and further introduces deformable convolution to enhance the flexibility and accuracy of feature extraction and improve the network’s ability to perceive the local content of images.We propose a Dual-branch Attention (DBA) structure, mechanism that uses channel attention to assign different weights to different feature channels and category specific residual attention to enhance the model’s sensitivity to image content, enabling the model to customize its evaluation for specific content or scene types in an image.We conducted evaluations on six benchmark datasets, encompassing both authentic and synthetic distortions, and our findings confirm that our proposed method yields good performance across diverse datasets.

## 2. Related Work

### 2.1. FR-IQA

In recent decades, FR-IQA methods have matured significantly with the emergence of many influential algorithms. FR-IQA methods are categorized into spatial domain methods and transform domain methods. Common spatial domain methods include the Peak Signal-to-Noise Ratio (PSNR) and the Structural Similarity Index (SSIM) [3]. PSNR is widely used to evaluate the quality of compressed images relative to the original images. This method is simple to calculate and fast, making it favored in the field of video processing. However, PSNR fails to consider the specific meanings of image content, and two images with significant quality differences may calculate the same PSNR value. In contrast, SSIM takes into account the characteristics of the human visual system, evaluating image similarity through comparisons of luminance, contrast, and structural features. Transform domain methods also play an important role in IQA. Common algorithms in this domain include the Wavelet Transform [4,5] and Discrete Cosine Transform (DCT) [6].

### 2.2. RR-IQA

While the FR-IQA algorithm has yielded impressive results, many applications only have access to parts of reference images, necessitating the development of RR-IQA. RR-IQA aims to minimize the dependency on full reference images by leveraging only partial information. Wu et al. [7] divide images into ordered and unordered parts, calculate the information content of each part, and assess fidelity, ultimately combining results to obtain a predictive score. Liu et al. [8] evaluate image quality by predicting differences between the original and test images through sparse representation. Wang et al. [9] employ Asymmetric Generalized Gaussian Density (AGGD) to fit the marginal distribution of calculated coefficients, thereby predicting image quality. Zhu et al. [10] decompose the original and test images into subband images, then extract free energy features through sparse representation to ultimately obtain a predictive score.

### 2.3. NR-IQA

In practical scenarios, evaluating image quality without reference images presents significant challenges. Unlike humans, computers require sophisticated algorithms to perform this task. NR-IQA algorithms are pivotal in these situations. Based on whether the distortion type is known, NR-IQA algorithms can be divided into specific distortion algorithms and general-purpose algorithms. Distortion-specific algorithms are designed to detect particular types of image distortions such as JPEG compression, blur, and various forms of noise. For instance, methods exist to evaluate JPEG compression by detecting the intensity of block effects [11] or intra-block blur effects [12]. Similarly, image blur can be assessed through edge analysis methods proposed by Marziliano et al. [13], or using transform domain methods like those employing DCT [14] or Discrete Wavelet Transform (DWT) [15]. Additionally, NR-IQA can estimate image noise using filter-based and other transform domain methods.

Specific distortion algorithms rely on prior assumptions about the type of distortion. However, in practical applications, images are often affected by multiple types of distortions, and it is difficult to determine the specific type of distortion, which poses a challenge for specific distortion algorithms to assess image quality. In contrast, general-purpose algorithms do not require assumptions about the type of distortion, but instead transform the no-reference image quality assessment problem into a classification or regression task, using extracted specific features for training. These relevant features can be extracted through Natural Scene Statistics (NSS) or identified via machine learning and deep learning techniques. Notable methods include BIQA [16], DIIVINE [17], BLIINDS [18], BRISQUE [19], and NIQE [20], which use NSS to model the statistical properties of distorted scenes to estimate image distortion levels.

With the advancement of machine learning techniques, learning-based methods have been proposed, with early research focusing on codebook learning methods. For example, Ye et al. [21] applied the K-means clustering algorithm to construct an unlabeled codebook from whitened original image blocks, used to map new image blocks to predict their quality scores. Xue et al. [22] extracted features of local blocks by segmenting and pooling distorted images, then used quality-aware clustering to identify cluster centers with different quality levels as codebooks; finally, the similarity between image blocks and the codebook centers was compared to assess image quality. Subsequently, sparse coding techniques were also applied in the field of no-reference image quality assessment.

With the development of CNN, the field of NR-IQA has also made significant progress. Kang et al. [23] utilized a five-layer CNN for accurate NR-IQA predictions, and in another study [24], developed a multi-task CNN that simultaneously estimates image quality and identifies distortion types. Zhang et al. [25] employed two branches to extract features from artificial and natural distortions separately, then fused these features through bilinear pooling to predict image quality. Furthermore, Ma et al. [26] proposed a multi-task, end-to-end optimized deep neural network, divided into two parts: the first for detecting distortion types and the second for predicting quality scores. Lin et al. [27] introduced generative adversarial networks to repair distorted images and generate pseudo-reference images, then predicted quality by comparing the distorted images with the pseudo-reference images, effectively compensating for the lack of reference in no-reference image assessments. Varga [28] implemented multiple CNN networks to simultaneously predict and weight image scores. Liu et al. [29] addressed the challenge of limited image datasets with the RankIQA model, and Yang et al. [30] combined a super-resolution network with CNNs for enhanced image quality prediction. Lastly, Su et al. [31] developed a hyper net to optimize the learning of fully connected layers, improving final score accuracy.

Transformer was initially introduced for Natural Language Processing (NLP) by Vaswani et al. [32] and later adapted for use in image processing. Early applications involved converting pixels into word vector-like formats. Subsequently, Chen et al. [33] implemented transformers in image classification tasks. J. You et al. [34] enhanced quality prediction by feeding CNN output features into transformers. S. Alireza Golestaneh et al. [35] integrated multi-scale information from CNNs into transformers for improved processing. Dosovitskiy et al. [36] introduced the Visual Transformer (ViT), which applies transformers directly across whole images. Junjie Ke et al. [37] further applied ViT to image quality evaluation, utilizing a multi-scale image quality converter to process images of varying resolutions and aspect ratios. Yang et al. [38] proposed a Multi-dimension Attention Network for No-Reference Image Quality Assessment, which utilizes ViT to extract image features. It integrates transpose attention blocks and scale-rotation Transformers to enhance global and local interactions between different regions of the image, thus effectively assessing image quality.

## 3. Proposed Methods

The proposed STNS-IQA is shown in Figure 1, which includes a deep learning branch and a NSS branch. We first introduce STNS-IQA pipeline and then discuss the details of our method.

### 3.1. Overview

Image quality assessment maps the input image to a quality score, a process that can be represented by the following equation:(1)f(I,θ)=q
where *f* denotes the network, I is the input image, θ corresponds to the learnable weight parameters, and *q* indicates the quality score.

The deep learning branch initially processes the image to be evaluated through resizing, cropping, and data augmentation to produce the input image $I\in R^{3\times H\times W}$, where *H*, *W* represent the height and width of the input image, respectively. Let $P_{i}\in R^{B\times C_{i}\times H_{i}\times W_{i}}$ denotes features exacted form the *i*th block of Swin-Transformer, where *B* and *C* means batch and channel, i∈{1,2,3,4}. Swin-Transformers use shifted window-based self-attention, which allows them to capture both fine-grained local interactions and, through the shifting mechanism, broader contextual relationships. Even on a smaller 14 × 14 feature map, this design enables effective modeling of both local and more global patterns. This is because each layer’s shifting windows integrate information from neighboring windows, effectively increasing the receptive field as you stack more layers. In addition, the hierarchical nature of Swin-Transformers allows for effective global information modeling. Through the use of multiple layers where windowing parameters and dilation can be adjusted, the network can learn to abstract higher-level features and contextual relationships, even from smaller maps. Multi-scale features are then fed into the Feature Enhancement Module (FEM) for processing to obtain $F_{i}$. Since the features extracted from different scales have varying channel numbers, heights, and widths, they are first standardized through a normalization layer that utilizes the Euclidean norm:(2)$${F_{i}}^{\prime }=\frac{F_{i}}{(max\left({∥F_{i}∥}_{2},\varepsilon \right)}$$
where ${F_{i}}^{\prime }$ represents the normalized feature map, and ε is a small quantity added to prevent division by zero in denominator. Then, the adaptive pooling layer down samples the features to $\bar{F_{i}}\in R^{B\times C_{i}\times H_{4}\times W_{4}}$ using $l_{2}$ pooling. The dropout layer is then used to reduce overfitting. Finally, we concatenate $\bar{F_{i}}$ for i∈{1,2,3,4} and obtain the output $F\in R^{B\times \sum_{i=1}^{4}C_{i}\times H_{4}\times W_{4}}$. Deformable convolution enhances model attention to detail and overall robustness by adapting convolution kernels to object shapes and sizes within images. Dual-branch attention prioritizes key features and amplifies critical information transmission, significantly boosting accuracy and efficiency in image quality assessment. NSS branch captures features at two scales (entire image and half size) processed through two fully connected layers (FC) to strengthen nonlinear relationships. Finally, features outputted by both deep learning and NSS branches are integrated using fully connected layers to compute the final image quality score.

In the deep learning branch, the Swin-Transformer convolutional layers use the GELU activation function for its smooth nonlinearity and better gradient flow, which enhance training stability and performance, the remaining convolutional layers employ ReLU activation functions to enhance the model’s convergence speed. In the natural scene statistics branch, PReLU activation functions are used between two fully connected layers to strengthen the nonlinear relationships between layers and expedite the convergence process. ReLU activation functions are also utilized in the final quality prediction stage. Since activation functions have already been introduced between convolutional layers, they are not applied after pooling operations to avoid unnecessary complexity.

### 3.2. Natural Scene Statistics Branch

The NSS branch employs BRISQUE-based features to analyze and assess image quality. At the initial stage of processing, we first compute the Mean Subtracted Contrast Normalized (MSCN) coefficients locally. For high-quality images, these coefficients typically conform to a Gaussian distribution, whereas those from distorted images deviate from this statistical norm. We use the generalized Gaussian distribution (GGD) to quantify this deviation, effectively capturing and extracting these regular deviations as features. Additionally, the product of neighboring MSCN coefficients also adheres to the statistical laws of natural images. By using AGGD, we are able to capture deviations in these product coefficients, thereby enhancing the robustness of feature extraction.

Specifically, we extract NSS features at scales of 1 and 1/2, and input them into two fully connected layers that use PReLU as the activation function, ultimately producing the output feature map H. By extracting features at multiple scales, the NSS method can effectively compensate for information loss caused by image scaling. Multiscale analysis allows us to capture image statistical features that remain consistent across different resolutions, ensuring that these features still effectively reflect the true quality of the image even after changes in image size.

### 3.3. Feature Enhancement Module

The Window Multiple Attention mechanism in the Swin-Transformer primarily focuses on features within a single small window. While the window shift strategy extends this focus beyond a single window by incorporating information from neighboring windows, it still does not achieve the broad context observability of a global attention mechanism. Inspired by advancements like Atrous Spatial Pyramid Pooling (ASPP) [39] and Shuffle Attention [40], our proposed Feature Enhancement Module (FEM) structure.

As shown in Figure 2, for the output of Swin-Transformer $P_{i}\in R^{B\times C_{i}\times H_{i}\times W_{i}}$, FEM utilizes channel shuffling and grouping convolution to classify the input feature maps into *G* groups, $P_{i}\in \left\{P_{i,1},\ldots ,P_{i,G}\right\}$, where $P_{i,k}\in R^{C_{i}/G\times H\times W}$. After grouping, the feature maps are processed through one convolutional layer, three dilated convolutional layers, and one pooling layer to produce five feature maps with the same number of channels. The pooling operation uses adaptive average pooling to obtain a feature map of size 1 × 1, which is then restored to the original spatial dimensions through bilinear interpolation. Given the diverse dimensions of output feature maps produced by the Swin-Transformer, we have strategically selected dilated convolutions with dilation rates tailored to 1/2, 1/4, and 1/8 of the respective feature map dimensions. This approach is designed to effectively capture and enhance the global features pertinent to each distinct scale. By adapting the dilation rates to match the varying scales of the feature maps, we optimize the receptive field for each layer, ensuring more precise and robust feature extraction across different levels of granularity. Each additional layer of dilated convolution expands the receptive field exponentially without loss of resolution or coverage. Three layers can provide a substantial increase in receptive field while maintaining a manageable increase in computational complexity. With three different scales (e.g., 1/2, 1/4, 1/8), the model is likely tuned to effectively capture features at multiple scales. Two layers could not offer sufficient scale diversity, whereas four layers bring more number of parameters and computational burden, and the performance will not be significantly improved and there is a risk of overfitting. These five feature maps are then concatenated to form the output ${P_{i,k}}^{\prime }\in R^{5C_{i}/G\times H\times W}$. Subsequently, ${P_{i,k}}^{\prime }$ is linearly mapped through the linear layer to obtain the output $F_{i,k}$ with the same number of channels as $P_{i,k}$. Finally, aggregation $F_{i,k}$ yields the final output $F_{i}$ of the FEM.

Compared to standard convolution, dilated convolution expands the receptive field without increasing the number of parameters, thus providing richer contextual information. As the Swin-Transformer produces multi-scale feature maps of different sizes, dilated convolution utilizes varying dilation rates tailored to each scale to maximize the receptive field. Grouped convolution, on the other hand, contributes to reducing the overall parameter count. Furthermore, the channel shuffling technique enhances the fusion and communication of information between different groups by rearranging the channel order within the feature maps. By integrating these methods, the FEM structure is optimized to process global information more effectively and to facilitate robust information exchange across various groups.

### 3.4. Deformable Convolution

The convolution kernel of standard convolution are fixed in the spatial sampling location, which limits the flexibility and adaptability of the model when dealing with objects of different sizes and shapes. For deep network structures that need to accurately capture and encode object size and shape information, this fixed sampling strategy makes it difficult to cope with dynamic changes in the image. In contrast, deformable convolution improves the model’s adaptability and ability to capture local feature changes by introducing offsets at the sampling locations and giving model the ability to automatically adjust the sensory field, allowing the convolution operation to more naturally adapt to the actual shape and size of the object. Applying deformable convolution to the field of image quality assessment allows the model to focus more accurately on the region of interest to the human eye, thus improving the accuracy of the assessment. Deformable convolution mesh sampling is illustrated in Figure 3, where (a) depicts a standard convolution operation, and (b), (c), and (d) demonstrate deformable convolutions. (c) and (d) are special cases of (b), representing how deformable convolution extends various transformations of scale, aspect ratios, and rotations. Through deformable convolution, F is transformed into output Z.

### 3.5. Dual-Branch Attention Mechanism

We propose a dual-branch attention architecture that integrates Category-Specific Residual Attention (CSRA) [41] and Channel Attention (CA) [42]. This architecture offers an effective method for enhancing the model’s ability to discern various image contents and detect quality defects, as shown in Figure 4. Firstly, CSRA can focus on enhancing the recognition capabilities for specific image categories. Through this approach, the model can not only identify the most important features within a category, but can also perform more precise quality assessments based on the content or type of scene in the images. This category-specific attention makes the model more efficient in processing images, as it emphasizes features that significantly impact the category while ignoring or suppressing information that is not important in the current category. Secondly, CA refines the model’s processing ability by assessing and amplifying the importance of different feature channels. This channel attention can identify which channels contain crucial information and reduce interference from irrelevant data by suppressing channels that carry less information or noise. Such fine control can significantly enhance the model’s ability to capture details, making it more sensitive and accurate in quality assessment. Finally, the combination of CSRA and CA allows the model to work synergistically on both spatial and channel dimensions, analyzing image quality at a finer granularity. This means it can parse each detail and potential flaw in the images in more detail.

For the output feature map $Z\in R^{C\times H\times W}$ resulting from deformable convolution, Channel Attention (CA) is initially applied to redistribute the weights of the feature map channels, yielding output $Z^{\prime }$. Subsequently, spatial pooling and average pooling are performed on $Z^{\prime }$ to produce outputs A and G, respectively. The results of these pooling operations are then weighted and combined to generate the final output S. Spatial pooling is weighted using the softmax function, which assigns greater weights to larger feature values while relatively suppressing smaller ones. This results in a better focus on salient feature information:(3)$$a_{i}=\sum_{j=1}^{HW}softmax{\left(Tz\right)}_{j}z_{j}$$
where *z* represents the feature map, *i* indicates the channel, *j* specifies the position within the feature map, and *T* is a modulation factor that determines the emphasis placed on the feature values at individual positions. For average pooling, a traditional approach is taken to compute the global information of the feature map:(4)$$g_{i}=\frac{1}{HW}\sum_{j=1}^{HW}z_{j}$$
the final feature map is computed by using the average pooling result as the main feature and the spatial pooling result as the residual feature:(5)$$s_{i}=g_{i}+\lambda a_{i}$$
where λ is an adjustment coefficient, controlling the impact of spatial feature information on the output results. In this paper, *T* is set to 2 and λ is set to 0.3.

### 3.6. Feature Fusion and Quality Prediction

The model integrates the output features from the deep learning branch and the natural scene statistics branch to predict image quality. To enhance accuracy and stability, normalization is applied to the ground truth and predict score prior to prediction. Introducing a normalized loss function stabilizes the gradient and increases predictability, thereby expediting the convergence of the IQA network. This approach significantly enhances the efficiency and accuracy of quality assessment. The process is represented by the following equation:(6)$$\left\{\begin{array}{c} \bar{q}=\sum_{i=1}^{m}q_{i}\\ {q_{i}}^{\prime }={\left({\sum_{i=1}^{m}\left|q_{i}-\bar{q}\right|}^{2}\right)}^{\frac{1}{2}} \end{array}\right.$$
(7)$$\left\{\begin{array}{c} \bar{s}=\sum_{i=1}^{m}s_{i}\\ {s_{i}}^{\prime }={\left({\sum_{i=1}^{m}\left|s_{i}-\bar{s}\right|}^{2}\right)}^{\frac{1}{2}} \end{array}\right.$$
where *m* is the number of images, $q_{i}$, $s_{i}$ represent predicted quality score and ground truth for the $i^{th}$ image, $q_{i}^{\prime }$, $s_{i}^{\prime }$ denote the normalized $q_{i}$, $s_{i}$, and $\bar{q}$, $\bar{s}$, respectively, represent the average predicted score of $q_{i}$, $s_{i}$. Finally, our loss function using $l_{2}$ norm is defined as:(8)$$L_{loss}=\frac{1}{N}\sum_{i=1}^{N}∥q_{i}^{\prime }-s_{i}^{\prime }∥$$

## 4. Experiment

### 4.1. Datasets and Evaluation Metrics

We evaluate our model on four synthetic distortion (LIVE [43], CSIQ [44], TID2013 [45], KADID-10K [46]) and two authentic distortion datasets (CLIVE [47], KonIQ-10K [48]). Table 1 shows the summary of the datasets that are used in our experiments.

We employed two general criteria, Spearman’s rank-order correlation coefficient (SROCC) and Pearson’s linear correlation coefficient (PLCC) for performance evaluation. SROCC is used to measure the monotonicity of predicted scores, i.e., whether the ranking of the predictions aligns with the ranking of the ground truth scores, reflecting the sensitivity of the IQA algorithm to variations in distortion levels, with values ranging from −1 to 1. When the SROCC value approaches 1, it indicates a strong positive correlation between the two datasets, suggesting good performance of the IQA (Image Quality Assessment) model. The formula for calculating SROCC is as follows:(9)$$SROCC=1-\frac{6\sum_{i=1}^{n}d_{i}^{2}}{D(D^{2}-1)}$$
where *D* denotes the number of samples and $d_{i}$ denotes the rank difference between the ground truth and predicted score for the $i^{th}$ image.

PLCC describes the linear correlation between two sets of data, with values ranging from −1 to 1, where a value closer to 1 indicates a stronger linear relationship between predicted score and ground truth. The formula for calculating PLCC is as follows:(10)$$PLCC=\frac{\left(s_{i}-\bar{s}\right)\left(q_{i}-\bar{q}\right)}{\sqrt{\sum_{i=1}^{m}{\left(s_{i}-\bar{s}\right)}^{2}}\sqrt{\sum_{i=1}^{m}{\left(q_{i}-\bar{q}\right)}^{2}}}$$
where $q_{i}$, $s_{i}$ represent predicted quality score and ground truth for the $i^{th}$ image, and $\bar{q}$, $\bar{s}$, respectively, represent the average predicted score of $q_{i}$, $s_{i}$.

### 4.2. Implementation Specifics

We constructed our model on the Pytorch framework. During the experimental process, we trained and tested the model using a NVIDIA RTX2080Ti GPU. During the training phase, the deep learning branch randomly crops the image into 224 × 224 blocks as input. For each training sample, 10 such image blocks are selected, and the quality score of the original image is assigned to these blocks. The same strategy is employed during the testing phase, where 10 image patches are randomly selected from each test image, and the quality scores of these patches are calculated. The final score is determined by averaging these scores. Adhering to the customary approach used by current IQA algorithms, we divided each dataset into training and testing subsets using an 8:2 ratio. This process was repeated five times, each with a different seed, to ensure variability in data splits. During training stage, the batch size is set to 8. We employ a cosine annealing learning rate, setting the initial learning rate to $1\times 10^{-5}$, with the parameters $T_{max}$ and $eta_{min}$ set to 50 and 0. The epoch is set to 20. Using the Adam optimizer, the weight decay is set to $1\times 10^{-5}$. To enhance the network’s convergence speed and learning capability, the backbone network Swin-Transformer will be pretrained on ImageNet.

### 4.3. Performance Evaluation

Table 2 shows a comparison of overall performance of PLCC and SROCC on four synthetic distorted image datasets, while Table 3 shows the overall performance comparison between PLCC and SROCC on two authentic distortion image datasets. Experimental results demonstrate that the model exhibits outstanding performance across multiple datasets. It has achieved good results across both six datasets. Additionally, the model shows high predictive accuracy on the LIVE and CSIQ datasets, but has a performance degradation on the TID2013 dataset, possibly due to the diverse range of distortion types in the dataset that impose higher demands on model precision. Overall, the model exhibits exemplary proficiency in managing both synthetic and authentic distortions across datasets of varying sizes, presenting competitive results. The superior performance of the algorithm is attributed to two main factors: firstly, the architecture of the deep learning branch, which includes the Swin-Transformer, a feature enhancement module, and deformable convolutions, enables the model to effectively integrate both global information and local details concurrently. Secondly, the NSS branch effectively compensates for information loss due to image cropping, thereby enhancing the overall performance of the model.

To further validate the effectiveness of the proposed STNS-IQA, an independent samples *t*-test is employed to compare the performance of two methods. The SROCC values for each model across six databases serve as the input for the *t*-test. The results are displayed in Table 4, where the labels “−1”, “0”, and “1” signify that the row method is statistically worse, equivalent to, and better than the column method at a 95% confidence level, respectively. According to Table 4, it is evident that the proposed STNS-IQA outperforms the other five deep learning-based NR-IQA methods in comparison.

We further compared the complexity of our model with DISTS, TReS, and StairIQA. As shown in Table 5, the STNS-IQA model has 46.84 M parameters and a computational complexity of 5.53 GFlops. Compared to other methods, STNS-IQA achieves a significant improvement in prediction accuracy, with a computational complexity that is only 35% of TReS.

Additionally, we conducted a Group Maximum Differentiation (gMAD) competition [62] on the Waterloo Exploration Database [63]. The two main roles in the gMAD competition are the attacker and the defender. The defender selects image pairs and believes they have the same quality, while the attacker finds pairs that the defender thinks have the same quality but actually have the largest quality difference. If human observers can easily distinguish the image quality, the attacker wins; otherwise, the defender wins. We conducted a gMAD competition between the proposed STNS-IQA and TReS [35], with results shown in Figure 5. Figure 5a,b demonstrate that when STNS-IQA is the defender and TReS is the attacker, TReS did not find pairs with significantly different quality. STNS-IQA’s predictions at both low and high quality levels align closely with human perception. In Figure 5c,d, TReS considers the image pairs to have the same quality, while STNS-IQA predicts the top images to be of higher quality than the bottom images. Overall, STNS-IQA more accurately reflects image quality differences compared to TReS, which targets both synthetic and real distortions.

### 4.4. Visual Analysis

Figure 6 evaluates the learning capability of potential features by the model. We compared the scores of similar images and observed that our model does not make assumptions about the type of distortion or the content of the pictures, yet it provides approximate scores to pictures with similar perceptual quality. This indicates that our model successfully learns the deep features of the pictures. Specifically, we selected two groups of images with neighboring prediction scores and provided objective scores in the test set of KonIQ-10K, with each group containing four images. The first row of Figure 6 depicts clear and bright images with high scores, while the second row shows poorly illuminated images with low scores. It can be observed that our model assigns close subjective scores to images with similar perceptual quality.

Weighted Gradient Class Activation Mapping (Grad-CAM) [64] is a powerful visual interpretation tool that utilizes gradient information within the model to assign weights to various channels of feature maps, thereby generating heat maps reflecting the regions of interest that the model focuses on. It can display the areas of interest that the model attends to. Grad-CAM technology was employed to generate attention heat maps on the KonIQ dataset to verify the consistency between the model-identified regions of interest and human interest areas. Through these heat maps, the focus points of the model when processing different images can be intuitively observed, as shown in Figure 7.

### 4.5. Ablation Study

In Table 6, the algorithm’s generalization is assessed through cross-database tests and compared with four other methods. The results indicate that training on a relatively large dataset leads to improved cross-dataset performance. Notably, when trained on the LIVE dataset and tested on the CSIQ dataset, favorable results were obtained, possibly due to similarities in features and distortion types between the two synthetic distortion datasets. Moreover, when the model was trained on the KonIQ dataset and tested on the CLIVE dataset, its transfer performance surpassed that of training on CLIVE and testing on KonIQ. Specifically, when trained on CLIVE and tested on KonIQ, the HyperIQA model achieved an SROCC of 0.772, slightly outperforming STNS-IQA’s 0.758. This difference may be attributed to the lightweight backbone model of ResNet50 employed in HyperIQA, facilitating easier convergence to a good point, especially with a smaller training dataset like CLIVE.In summary, the STNS-IQA model demonstrates strong generalization ability.

To verify the validity of our proposed model, we conducted ablation experiments and illustrated the effect of each part of the model on KonIQ-10K dataset, the experimental results are shown in Table 7.

We further verify the effectiveness of the Normalized Mean Squared Error (NMSE) loss function used in this chapter, it was compared with other classical loss functions, Mean Absolute Error (MAE) and Mean Squared Error (MSE), on the KonIQ-10K dataset. The comparison results, as shown in Figure 8, clearly demonstrate that NMSE is superior in terms of convergence speed and stability during training compared to MAE and MSE. This indicates that NMSE provides a faster and more reliable optimization path, thereby significantly enhancing the model performance and stability for the task of image quality assessment.

As shown in Figure 9, we have demonstrated the changes in the training and validation loss curves on the KonIQ-10K dataset. From the figure, it can be observed that throughout the training process, both the training and validation losses decrease, with a significant drop in the first five epochs, followed by a more gradual decline. After the 15th epoch, the curves stabilize, indicating that the model is learning and progressively optimizing. Additionally, it is evident that the training loss curve is on a downward trend, while the validation loss curve does not show an upward trend, suggesting that the model is not overfitting.

## 5. Conclusions

We propose a NR-IQA model, named STNS-IQA, which combines Swin-Transformer and natural scene statistics. Swin-Transformer is utilized to extract multi-scale information from images. We introduce a feature enhancement module to gather more contextual information. We also incorporate deformable convolutional layers to enhance the model’s attention to global image content and objects. Natural scene statistics methods are employed to compensate for potential data loss caused by image resizing, aiding in assessing the visual naturalness and quality of images. A dual-branch attention mechanism is introduced to focus on key regions, and a normalized loss function is employed to improve the model’s convergence speed and stability. Compared to existing methods, STNS-IQA performs well on both synthetic and authentic distortion datasets, achieving optimal results on multiple datasets and demonstrating strong generalization capabilities. In future work, the use of lightweight networks is considered to reduce the number of parameters while maintaining high evaluation accuracy.

## Figures and Tables

**Figure 1 sensors-24-05221-f001:**
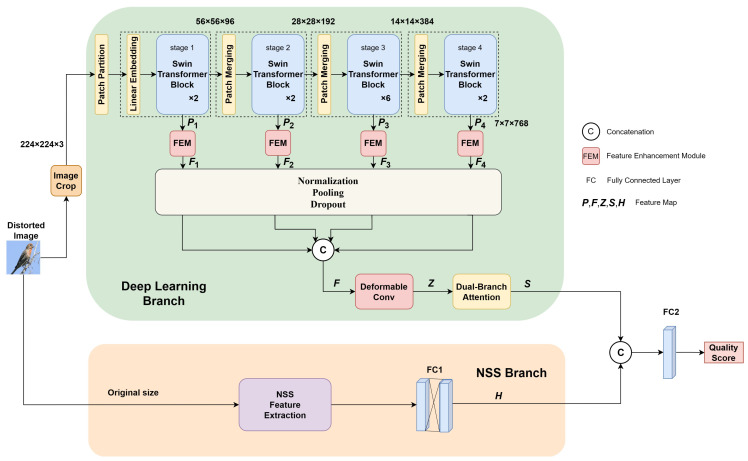
Framework of STNS-IQA.

**Figure 2 sensors-24-05221-f002:**
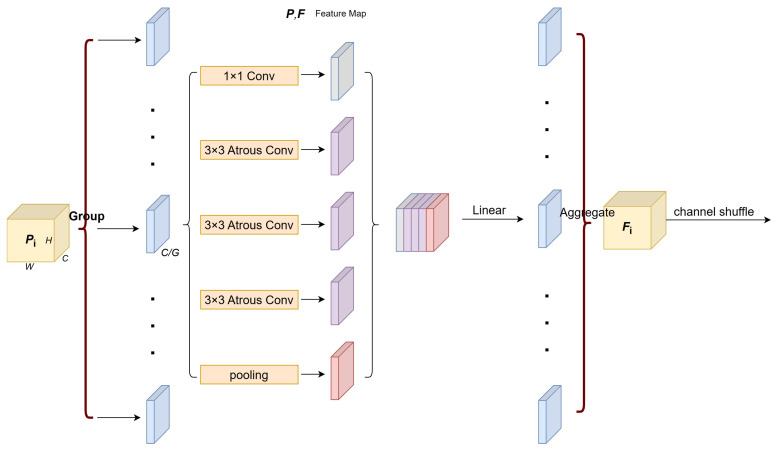
Artifact of feature enhancement module.

**Figure 3 sensors-24-05221-f003:**
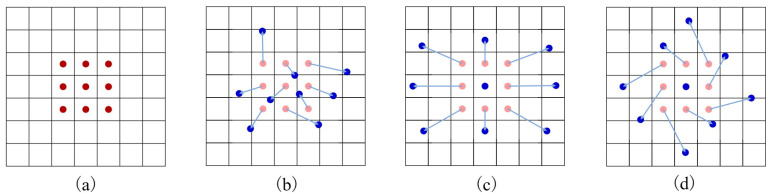
Deformable convolution mesh sampling. The red dot indicates the original position of the convolution kernel, the blue arrow represents the offset of the convolution kernel, and the blue dot indicates the position of the deformable convolution kernel. (**a**) standard convolution. (**b**–**d**) Different morphologies of deformable convolutions.

**Figure 4 sensors-24-05221-f004:**
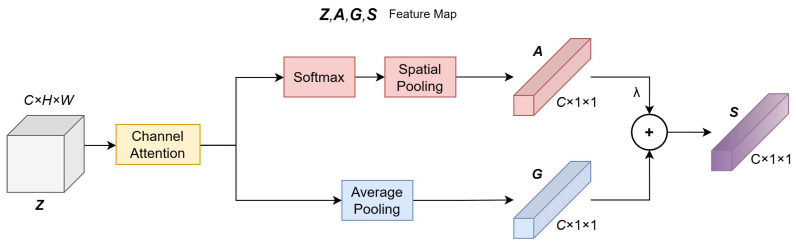
Artifact of dual-branch attention.

**Figure 5 sensors-24-05221-f005:**
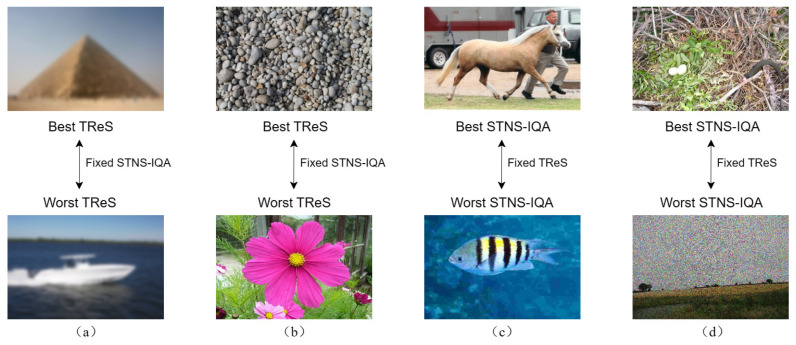
The gMAD competition results between TReS and STNS-IQA. (**a**,**b**) Best (top) and worst (bottom) images selected when STNS-IQA is the defender and TReS is the attacker. (**c**,**d**) Best (top) and worst (bottom) images selected when TReS is the defender and STNS-IQA is the attacker.

**Figure 6 sensors-24-05221-f006:**
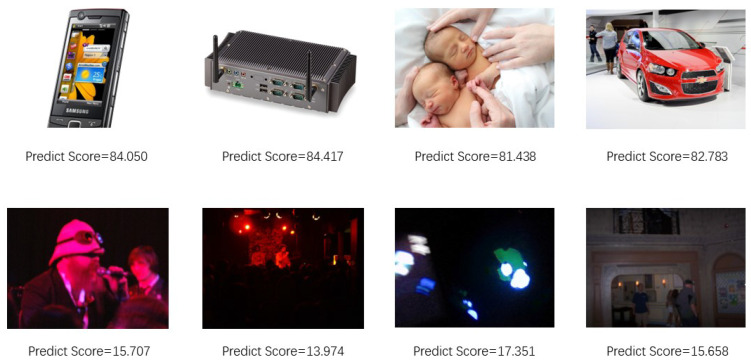
Images with similar Predict Scores.

**Figure 7 sensors-24-05221-f007:**
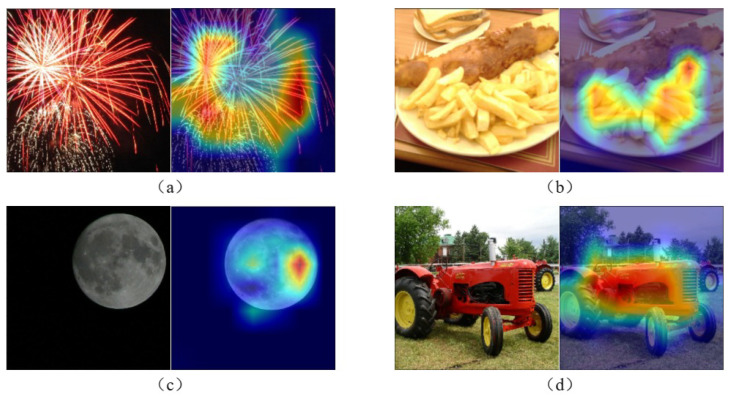
Original image (**left**) and Grad-CAM heat map (**right**). (**a**) primarily focuses on the fireworks area. The bright colors and strong contrast of the fireworks are key elements that attract attention, and the model effectively captures this, indicating its similarity to human vision in reacting to bright and dynamic objects. (**b**) The focus is concentrated on the food in the center of the image. This demonstrates the effectiveness of the model in processing objects with rich details and color variations. (**c**) The attention is directed towards the area of the moon, likely due to its brightness and prominent position in the night sky. (**d**) The tractor is the focus of attention, possibly because it is the main element in the image, with distinct structure and color differences. Through these results, we can observe that the model’s areas of focus approximate those of human eye’s initial perception, indicating that the model can effectively simulate human visual attention to different scenes.

**Figure 8 sensors-24-05221-f008:**
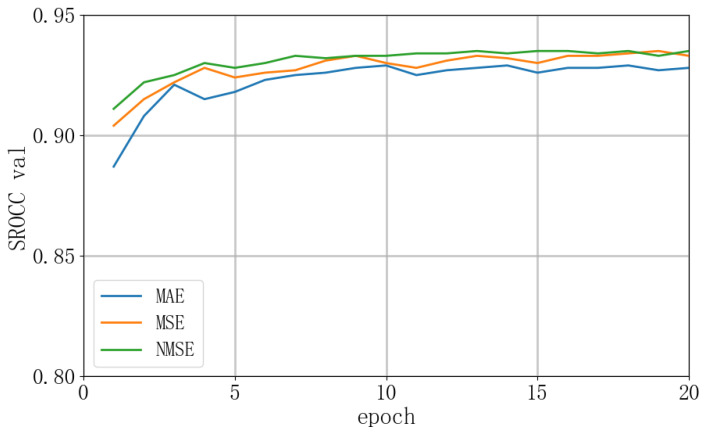
The validation curves on KonIQ-10K.

**Figure 9 sensors-24-05221-f009:**
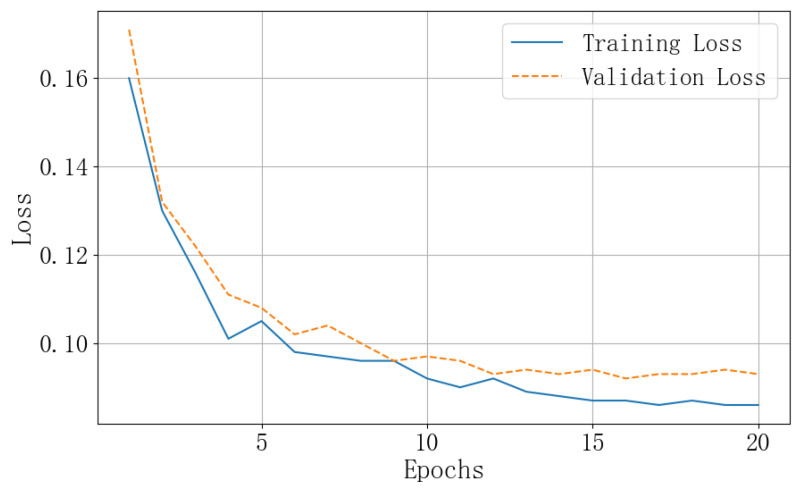
Loss function curves on KonIQ-10K dataset.

**Table 1 sensors-24-05221-t001:** Summary of IQA datasets.

Datasets	# of Dist. Images	# of Dist. Types	Distortions Type
LIVE	799	5	synthetic
CSIQ	866	6	synthetic
TID2013	3000	24	synthetic
KADID-10K	10,125	25	synthetic
CLIVE	1162	-	authentic
KonIQ-10K	10,073	-	authentic

**Table 2 sensors-24-05221-t002:** Result on synthetic distortion datasets. Blue and black numbers in bold represent the best and second best, respectively.

Methods	LIVE		CSIQ		TID2013		KADID-10K	
	SROCC	PLCC	SROCC	PLCC	SROCC	PLCC	SROCC	PLCC
DIIVINE [49]	0.892	0.908	0.804	0.776	0.643	0.567	0.413	0.435
BRISQUE [19]	0.929	0.944	0.812	0.748	0.626	0.571	0.528	0.567
ILNIQE [50]	0.902	0.906	0.822	0.865	0.521	0.648	0.528	0.558
BIECON [51]	0.958	0.961	0.815	0.823	0.717	0.762	0.623	0.648
MEON [26]	0.951	0.955	0.852	0.864	0.808	0.824	0.604	0.691
DBCNN [25]	0.968	0.971	0.946	0.959	0.816	0.865	0.851	0.856
TIQA [34]	0.949	0.965	0.825	0.838	0.846	0.858	0.850	0.855
MetaIQA [52]	0.960	0.959	0.899	0.908	0.856	0.868	0.762	0.775
P2P-BM [53]	0.959	0.958	0.899	0.902	0.862	0.856	0.840	0.849
HyperIQA [31]	0.962	0.966	0.923	0.942	0.840	0.858	0.852	0.845
TReS [35]	0.969	0.968	0.922	0.942	0.863	0.883	0.858	0.859
VCRNet [54]	0.971	0.972	0.943	0.955	0.846	0.875	-	-
MB-CNN [55]	0.972	0.972	0.937	0.949	0.808	0.842	-	-
DACNN [56]	**0.978**	**0.980**	0.943	0.957	0.871	0.889	0.905	0.905
MSTRIQ [57]	-	-	-	-	0.882	0.895	-	-
SaTQA [58]	-	-	**0.965**	**0.972**	**0.938**	**0.948**	**0.946**	**0.949**
**STNS-IQA**	**0.977**	**0.979**	**0.966**	**0.976**	**0.908**	**0.922**	**0.922**	**0.926**

**Table 3 sensors-24-05221-t003:** Result on authentic distortion datasets. Blue and black numbers in bold represent the best and second best, respectively.

Methods	CLIVE		KonIQ-10K	
	SROCC	PLCC	SROCC	PLCC
DIIVINE [49]	0.588	0.591	0.546	0.558
BRISQUE [19]	0.629	0.629	0.681	0.685
ILNIQE [50]	0.508	0.508	0.523	0.537
BIECON [51]	0.613	0.613	0.651	0.654
MEON [26]	0.697	0.710	0.611	0.628
DBCNN [25]	0.869	0.869	0.875	0.884
TIQA [34]	0.845	0.861	0.892	0.903
MetaIQA [52]	0.835	0.802	0.887	0.856
P2P-BM [53]	0.844	0.842	0.872	0.885
HyperIQA [31]	0.859	0.882	0.906	0.917
TReS [35]	0.846	0.877	0.915	0.928
VCRNet [54]	0.856	0.865	0.894	0.909
SCA-IQA [59]	-	-	**0.916**	**0.931**
DACNN [56]	**0.866**	0.884	0.901	0.912
MSTRIQ [57]	-	-	**0.946**	**0.954**
SaTQA [58]	**0.877**	**0.903**	**0.930**	**0.941**
**STNS-IQA**	0.864	**0.890**	0.923	0.933

**Table 4 sensors-24-05221-t004:** The *t*-test results of the different methods.

Method	DBCNN [25]	TIQA [34]	HyperIQA [31]	TReS [35]	DACNN [56]	STNS-IQA
DBCNN [25]	0	0	−1	−1	−1	−1
TIQA [34]	0	0	−1	−1	−1	−1
HyperIQA [31]	1	1	0	−1	−1	−1
TReS [35]	1	1	1	0	−1	−1
DACNN [56]	1	1	1	1	0	−1
STNS-IQA	1	1	1	1	1	0

**Table 5 sensors-24-05221-t005:** Comparison of SROCC results and complexity with competitive methods.

Method	GFlops	Params. (M)	CSIQ	TID2013	KonIQ-10K
DISTS [60]	30.69	14.72	0.929	0.830	-
TReS [35]	8.39	34.46	0.922	0.863	0.915
StairIQA [61]	10.38	31.80	0.919	-	0.921
STNS-IQA	5.53	46.84	0.966	0.908	0.923

**Table 6 sensors-24-05221-t006:** SROCC evaluations on cross datasets, where bold entries indicate the best performers.

Train On	CLIVE	KonIQ-10K	LIVE
Test On	KonIQ-10K	CLIVE	CSIQ
WaDIQaM [65]	0.711	0.682	0.704
DBCNN [25]	0.754	0.755	0.758
HyperIQA [31]	**0.772**	0.785	0.744
TReS [35]	0.733	0.786	0.761
**STNS-IQA**	0.758	**0.808**	**0.763**

**Table 7 sensors-24-05221-t007:** Ablation study results on KonIQ-10K.

Swin-Transformer	FEM	Deformable Convolution	DBA	NSS Branch	SROCC
✓					0.912
✓	✓				0.915
✓		✓			0.914
✓	✓	✓			0.919
✓	✓	✓	✓		0.921
✓	✓	✓	✓	✓	0.923

## Data Availability

Data are contained within the article.

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
