# Peer review of "No-Reference Image Quality Assessment Combining Swin-Transformer and Natural Scene Statistics"

_sensors, 2024, doi:10.3390/s24165221_

Round 1

Reviewer 1 Report

Comments and Suggestions for Authors

This paper proposes a no-reference image quality assessment model that combines Swin-Transformer and natural scene statistics.  It employs Swin-Transformer to extract multi-scale features from images, and natural scene statistics to compensate for the information loss caused by image resizing. Overall, the writing of the paper is standardized, the experiments are sufficient, and it has certain reference value, but there are still a few issues.

1. As shown in Figure 1,the size of the feature map has been reduced to 14 * 14. Does it still need to be windowed (applied Swin Transformer) for such a small size? Can it demonstrate the global modeling capability of Transformer?

2. Need to add and analyze some recent references, especially Swin Transformer in image quality evaluation, and add some experimental comparisons. 

[1] Transformer-based No-Reference Image Quality Assessment via Supervised Contrastive Learning,AAAI24

[2] MSTRIQ: No Reference Image Quality Assessment Based on Swin Transformer with Multi-Stage Fusion, CVPRW 22.

Author Response

Comments 1:

As shown in Figure 1,the size of the feature map has been reduced to 14 * 14. Does it still need to be windowed (applied Swin Transformer) for such a small size? Can it demonstrate the global modeling capability of Transformer?

Response 1:

Thank you very much for your suggestion and insightful comments. I have made modifications and loaded them into section 3.1 on page 5.

Using Swin Transformers on a reduced feature map size of 14x14 still makes sense depending of our model.

First, Swin Transformers use shifted window-based self-attention, which allows them to capture both fine-grained local interactions and, through the shifting mechanism, broader contextual relationships. Even on a smaller 14x14 feature map, this design enables effective modeling of both local and more global patterns. This is because each layer's shifting windows integrate information from neighboring windows, effectively increasing the receptive field as you stack more layers.

In addition, the hierarchical nature of Swin Transformers allows for effective global information modeling. Through the use of multiple layers where windowing parameters and dilation can be adjusted, the network can learn to abstract higher-level features and contextual relationships, even from smaller maps.

In summary, applying Swin Transformer to a 14x14 feature map can still effectively leverage the model’s capability to capture both local and global dependencies, It's a balanced approach between maintaining manageable computational costs while potentially enhancing the feature's representational power. Potentially enhances the expressiveness of features while maintaining a manageable computational cost.

Comments 2:

Need to add and analyze some recent references, especially Swin Transformer in image quality evaluation, and add some experimental comparisons.

[1] Transformer-based No-Reference Image Quality Assessment via Supervised Contrastive Learning,AAAI24

[2] MSTRIQ: No Reference Image Quality Assessment Based on Swin Transformer with Multi-Stage Fusion, CVPRW 22.

Response 2:

Thank you for your valuable suggestion. In response, I have incorporated the two methods you referenced into section 4.3 of the paper. 

Reviewer 2 Report

Comments and Suggestions for Authors

In this paper, authors proposed a model that combines Swin-Transformer, which is extracting multi-scale features, and Natural Scene Statistics, which is compensating information loss caused by image resizing in order to assess image quality without reference. The Manuscript is very well prepared, organized and clear, with confirmed accurate model prediction using Spearman rank correlation coefficients. However, it has minor flaws commented below.

Feature Enhancement Module (FEM) is developed to enhance global visual information capture using dilated and deformable convolution to ensure flexibility and accuracy of feature extraction. FEM is very important part of the model and some minor explanation is needed. The authors used 3 Atrous Convolution with size 3x3, without explaining how and why did they decide to use 3 Atrous Convolution. Did the research start with trial and error, and the authors concluded that 3 Atrous Convolution is the optimized model? Will more Atrous Convolution layers cause more accurate prediction or overfitting? Please add explanation.

In the FEM model, it is not mentioned the size of pooling, which is usually 2x2, but it is important to define its size.

It is not defined which activation function is used in each of the convolution and pooling. It’s a very big difference between linear and nonlinear activation functions, and it is of crucial importance to define them, in order to better understand the prediction process and the results.

Authors should define number of epochs in the training stage, since high number of epochs can lead to overfitting, it is worth mentioning and how it is affecting the results.

Author Response

Comments 1:

Feature Enhancement Module (FEM) is developed to enhance global visual information capture using dilated and deformable convolution to ensure flexibility and accuracy of feature extraction. FEM is very important part of the model and some minor explanation is needed. The authors used 3 Atrous Convolution with size 3x3, without explaining how and why did they decide to use 3 Atrous Convolution. Did the research start with trial and error, and the authors concluded that 3 Atrous Convolution is the optimized model? Will more Atrous Convolution layers cause more accurate prediction or overfitting? Please add explanation.

Response 1:

Thank you for your suggestion. I did not explain the part about atrous convolutions in the Feature Enhancement Module clearly. I have provided a more detailed explanation of atrous convolutions in Section 3.3, as shown below.

Given the diverse dimensions of output feature maps produced by the Swin-Transformer, we have strategically selected dilated convolutions with dilation rates tailored to 1/2, 1/4, and 1/8 of the respective feature map dimensions. This approach is designed to effectively capture and enhance the global features pertinent to each distinct scale. By adapting the dilation rates to match the varying scales of the feature maps, we optimize the receptive field for each layer, ensuring more precise and robust feature extraction across different levels of granularity.

Each additional layer of dilated convolution expands the receptive field exponentially without loss of resolution or coverage. Three layers can provide a substantial increase in receptive field while maintaining a manageable increase in computational complexity. With three different scales (e.g., 1/2, 1/4, 1/8), the model is likely tuned to effectively capture features at multiple scales. Two layers could not offer sufficient scale diversity, whereas four layers bring more number of parameters and computational burden, and the performance will not be significantly improved and there is a risk of overfitting.

Comments 2:

In the FEM model, it is not mentioned the size of pooling, which is usually 2x2, but it is important to define its size.

Response 2:

Thank you for your suggestion. I have trouble explaining the pooling operation clearly. I provided a more detailed explanation of the pooling operation in Section 3.3, as follows.

The pooling operation uses adaptive average pooling to obtain a feature map of size 1x1, which is then restored to the original spatial dimensions through bilinear interpolation.

Comments 3:

It is not defined which activation function is used in each of the convolution and pooling. It’s a very big difference between linear and nonlinear activation functions, and it is of crucial importance to define them, in order to better understand the prediction process and the results.

Response 3:

Thank you for your correction. I have added the explanation about activation functions to section 3.1 in the manuscript, as follows.

In the deep learning branch, the Swin-Transformer convolutional layers use the GELU activation function for its smooth nonlinearity and better gradient flow, which enhance training stability and performance, the remaining convolutional layers employ ReLU activation functions to enhance the model's convergence speed. In the natural scene statistics branch, PReLU activation functions are used between two fully connected layers to strengthen the nonlinear relationships between layers and expedite the convergence process. ReLU activation functions are also utilized in the final quality prediction stage. Since activation functions have already been introduced between convolutional layers, they are not applied after pooling operations to avoid unnecessary complexity.

Comments 4:

Authors should define number of epochs in the training stage, since high number of epochs can lead to overfitting, it is worth mentioning and how it is affecting the results.

Response 4:

Thank you for your suggestion. I neglected to mention earlier that the model's epochs were set to 20, which I have now added in Section 4.2.

In addition, in Section 4.5, I have illustrated the changes in the training and validation loss curves on the KonIQ-10K dataset, as shown below.

As shown in Figure 9, we have demonstrated the changes in the training and validation loss curves on the KonIQ-10K dataset. From the figure, it can be observed that throughout the training process, both the training and validation losses decrease, with a significant drop in the first 5 epochs, followed by a more gradual decline. After the 15th epoch, the curves stabilize, indicating that the model is learning and progressively optimizing. Additionally, it is evident that the training loss curve is on a downward trend, while the validation loss curve does not show an upward trend, suggesting that the model is not overfitting.
